# RDLNet: A Novel and Accurate Real-world Document Localization Method

## ABSTRACT

The increasing use of smartphones for capturing documents in various real-world conditions has underscored the need for robust document localization technologies. Current challenges in this domain include handling diverse document types, complex backgrounds, and varying photographic conditions such as low contrast and occlusion. However, there currently are no publicly available datasets containing these complex scenarios and few methods demonstrate their capabilities on these complex scenes. To address these issues, we create a new comprehensive real-world document localization benchmark dataset which contains the complex scenarios mentioned above and propose a novel Real-world Document Localization Network (RDLNet) for locating targeted documents in the wild. The RDLNet consists of an innovative light-SAM encoder and a masked attention decoder. Utilizing light-SAM encoder, the RDLNet transfers the mighty generalization capability of SAM to the document localization task. In the decoding stage, the RDLNet exploits the masked attention and object query method to efficiently output the triple-branch predictions consisting of corner point coordinates, instance-level segmentation area and categories of different documents without extra post-processing. We compare the performance of RDLNet with other state-of-the-art approaches for real-world document localization on multiple benchmarks, the results of which reveal that the RDLNet remarkably outperforms contemporary methods, demonstrating its superiority in terms of both accuracy and practicability.

## CCS CONCEPTS

• **Applied computing → Document scanning**.

## KEYWORDS

Document Localization, Novel Benchmark Dataset, Encoder and Decoder Based Network, Distillation, Triple Branch Prediction

## 1 INTRODUCTION

In recent decades, there has been an increasing number of users starting to use mobile devices to take photos or videos of the paper documents. Document digitization has become a trend in our society [40]. Once documents are digitized, they will be easier to store, retrieve, and read. To achieve this, the document must first be accurately located in the image. Therefore, document localization plays a crucial role in the document digitization process. However, localizing documents in the natural scene images still faces many challenges in practical applications. Firstly, images can be taken from different distance and viewpoints, the documents in the image will have various sizes, positions, and perspective distortions. Additionally, in some cases, images are usually taken in a casual way. For instance, some captured images are often blurry, possessing different lighting conditions, low contrast, some occlusion in the document area and even similar overlapping background documents. Therefore, document localization in real world images remains a challenging problem.

Although many advanced methods have achieved good results in the field of document localization, experimental verifications of most methods are based on existing opensource document detection datasets, such as SmartDoc dataset [4] and CDPhotoDataset [11]. But these datasets usually contains limited document types. Moreover, the pictures in these datasets are taken in the scenes which are relatively simple and do not contain comprehensive real-world situations. These complex situations are common in practical deployments and inevitable. However, to the best of our latest knowledge, there is not a comprehensive dataset that fully reflects complex scenarios of document images in the wild. Therefore, many methods have reached near-saturation localization accuracy on current datasets, but this does not fully prove that the challenges existing in the real-world have been well solved. That is why a more extensive and comprehensive real-world document dataset is needed to provide a more valuable and research-space benchmark for future research.

As mentioned above, many previous advanced studies have achieved excellent results in the field of document detection, but they all focused on a single semantic-level document type and simple photography scenarios. And these methods do not fully demonstrate their generalization performance. In recent years, the emergence of large models has revolutionized the applications in many fields. In the application of image segmentation, the emergence of SAM [18] unifies the tasks of instance segmentation and semantic segmentation and shows superior performance. However, SAM is a segmentation model for general image scenes and can be used for document edge detection rely on its prompt design. Moreover, due to the large scale of the SAM model, it is not suitable for running on mobile devices such as personal computers or smartphones.

To solve the above problems, we propose a novel Real-world Document Localization Network (RDLNet) and create a new complex and comprehensive benchmark dataset which is Real world Mobile Document dataset (RWMD) captured by mobile phones in the wild. First, in order to better transfer the powerful generalization ability of SAM in segmentation tasks to document localization tasks, we design a lightweight SAM encoder module. Second, after the document image is encoded by the light-SAM, we propose an advanced decoder module based on masked attention and object query. Third,

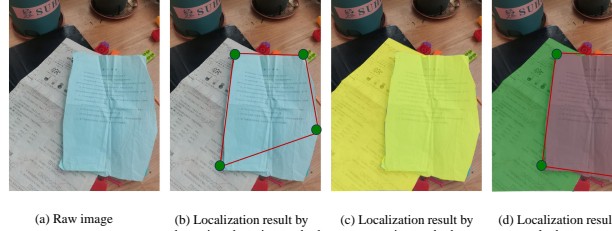

(a) Raw image

(b) Localization result by keypoints detection method

(c) Localization result by segmentation method

(d) Localization result by our method

**Figure 1: Document localization samples of different methods.**

we design three output branches based on the features from both the encoder and decoder modules. The three branches contain the classification branch, mask branch and corner points branch. Different from the previous methods of corner prediction based directly on decoded semantic features, we design an innovative branch based on the hybrid information of semantic features, mask features and object query vectors. Hence, the accurate document localization results can be directly obtained by the corner prediction branch. Document localization samples of different methods are presented in Figure 1. Specially, our method can simultaneously output corner point predictions of foreground document and region prediction results that distinguish among different documents in one image. Finally, through the above proposed modules, our model's able to generate accurate document localization results with high efficiency, especially on the new comprehensive benchmark dataset we contributed with more distinct effect.

In summary, the main contributions of this paper includes:

(1). **Light-SAM encoder**: We introduce the light-SAM encoding module into the field of document localization, and exploit distillation method to constrain and reduce the parameters of the designed encoding module to improve model performance.

(2). **Masked attention and object query based decoder**: We design an advanced decoding module based upon masked attention and object query to allow the model to form the global and local relations between document objects and image context to directly and separately output the predictions of different documents in one image or frame.

(3). **Triple-branch**: We propose a novel triple-branch that integrates hybrid features to predict corner points, area masks and categories at the same time. By jointly optimizing the losses of the three branches, we can get more accurate prediction results. In addition, the number of corner points of the document can be modified as a hyper-parameter in our method.

(4). **New benchmark dataset**: We create a new benchmark dataset which is called Real world Mobile Document(RWMD) dataset for document localization. This dataset provides rich document types and diverse shooting scenarios in the wild. RWMD fills the current lack of comprehensive challenging document localization datasets. And the RWMD dataset will be soon publicly available.

(5). **SOTA results:** In the experiments, we compare the performance of our method and other current state-of-art methods in document localization on multiple datasets contain our new

dataset RWMD. The results proved that the accuracy of our proposed method exceeded other methods with low inference time.

## 2 RELATED WORK

### 2.1 Document Localization Datasets

There are several public datasets used to measure the capabilities of document localization methods: SmartDoc dataset [4], Extended SmartDoc dataset [10], CDPhotoDataset [11], MIDV dataset family (MIDV-500 dataset [1], MIDV-2019 dataset [2], MIDV-2020 [3]). SmartDoc dataset was created for ICDAR 2015 SmartDoc competition. Each of these documents are printed using a color laserjet and captured by a Google Nexus 7 tablet [4]. The database consists of 150 video clips comprising 25,000 frames. Many research works have been carried out based on this dataset [4, 8, 15, 34]. However, SmartDoc dataset only contains smooth A4 paper format, and most backgrounds are in close similarity and are lack of diversity. Ricardo, et al.[10] created an extension of the original Smartdoc Dataset known as the Extended SmartDoc dataset. The authors inserted extra backgrounds into the original images with backgrounds acquired from various smartphone cameras with richer scenes than the original datasets. This dataset has a total of 29,522 images with a fixed size of 1728 × 2304 pixels. However, the data synthesis method of this dataset doesn't sufficiently increase the richness of the original dataset. The CDPhotoDataset [11] is composed of images of Brazilian identification documents, in different backgrounds of real-world environments, with non-uniform standards. It has a total of 20,000 images of documents with the respective document mask as ground truth. Since the document images in this dataset are manualy merged into background images, this dataset cannot fully reflect the real-world document photo scene. The first dataset of the MIDV family is MIDV-500 [1] which contains 500 video clips of 50 identity documents, 10 clips per document type. The conditions represented in MIDV-500 thus have some diversity regarding the background and the positioning of the document, however, they don't include variation in lighting conditions, or significant projective distortions. To address the issues the dataset MIDV-2019 [2] was later published as an extension of MIDV-500, which featured video clips captured with very low lighting conditions and with higher projective distortions. The main disadvantage of the datasets of the MIDV-2019 is the scarcity of different document samples. All images in this dataset are made using the same 50 physical document samples. In order to make up for this shortcoming, MIDV2020 [3] photo dataset consists of 1000 different physical documents, all with unique artificially generated faces, signatures, and text fields data. Although this dataset contains some complex scenes, it only contains hard cards and therefore does not include some shape changes and flexible transformations.

### 2.2 Document Localization Methods

Research has been conducted on document localization since the start of camera-based document processing development [17]. The earliest methods were primarily designed to solve scanned documents. These documents are required to be placed flatly in a prescribed area, so the background is simple. Then the traditional segmentation methods[20, 27, 30] can locate the document easily. In the recent literature addressing document localization problems,

these methods can be broadly divided into four categories: document boundary-detection-based methods, segmentation methods, keypoints detection methods, document content-based methods.

**Document boundary-detection-based methods**: These methods continue the idea of traditional image processing methods and locates documents by detecting the edges of the document. Document edge detection methods usually include the following methods: edge extraction methods (such as Canny detector, Sobel operator, or Holistically-Nested Edge Detector [36]), line detectors (like Line Segment detectors [13], Hough transform). This type of methods is robust to corner occlusion, but requires a clean background and clear document edges. Document detection accuracy heavily depends on environmental conditions and shooting situations. Once the conditions change, such as low contrast and light intensity, the accuracy of document localization results will be significantly affected.

**Segmentation methods**: These methods follow the flow of object segmentation methods [14, 21, 28, 32, 35, 39], which treat the document as an object. This type of methods [7, 8, 37] uses dense prediction to output a classification result for each pixel to distinguish whether the pixel belongs to the document or the background. Then by grouping the pixels with the same label, the document is segmented. This type of method is robust to irregular documents. But these methods require further post-processing to get the orientation of the document. Moreover, these methods usually have large model sizes and require more inference time, making it difficult to meet the efficiency requirements for processing on mobile devices.

**Keypoints detection methods**: This type of methods [15, 26, 40] usually assumes that the document has four corner points, and then converts the document localization problem into a corner positioning problem. It can be solved using corner detectors and key point detectors. If the used dataset has corner annotations, this type of methods can directly obtain the orientation of the document based on the predicted corner points coordinates. However, these methods will be ineffective for documents whose corners are occluded or have irregular shapes.

**Document content-based methods**: This type of methods [23, 24, 31] transforms the document localization problem into the positioning problem of the text content area. These methods use the layout and pattern characteristics within the document to locate document areas, such as text line information, image or table edge information, and the layout of different text blocks. The advantages of this type of methods are that it can judge the orientation of the document based on the information of the text content. And it is conducive to extracting useful content information for subsequent recognition steps. However, these methods usually assume the background to be simple and flat in contrast to the document. Therefore, these methods often fail when the background is complex or confusing.

There are also some literature that use hybrid methods, such as [29] which uses both corner prediction and edge prediction to locate documents. And [34] combines corner points prediction, line boarders prediction and document classification in a single lightweight network. Our method belongs to this kind of category combining the corner points prediction, instance-level segmentation, and classification.

## 2.3 SAM

SAM [18] is a powerful model with strong generalization ability for segmentation tasks based upon user prompts. It's pre-trained on a broad dataset SA-1B consisting of 11 million images and 1 billion masks. Given a prompt like a single ambiguous point prompt on the document in an image, it's capable of generating the document's area mask, and with post-processing, corner points of that document can be obtained. Nonetheless, it's computationally costly since it contains at least 91 million parameters which is too large for mobile devices.

## 3 METHOD

The pipeline of the proposed RDLNet is depicted in Figure 2. In this section we elucidate distillation process we applied for the backbone, following the encoder-decoder structure we adopted and the loss functions for RDLNet's training supervision.

## 3.1 Light-SAM Backbone Multiplex Distillation

SAM [18] is a powerful model for generic and referring segmentation. It is based on the Vision Transformer(ViT) [12]. The ViT is a stack of Transformer blocks, which is used to extract features from the RGB input image $I \in \mathbb{R}^{H \times W \times 3}$ of size $H \times W$ with three channels.

Concretely, the input image $I$ is divided and flattened into a sequence of non-overlapping patches $I' \in \mathbb{R}^{N \times C}$, where $C = P^2 \times 3$ and $N = \frac{HW}{P^2}$, assuming the patch size is $P \times P$. They are then linearly projected to obtain a sequence of $C'$-dimension feature vectors, which are fed into the Transformer encoder with each block ($l$) consisting of multi-head self-attention (MSA) and feedfoward network(FFN) conducting operations below:

$$\begin{aligned} \mathbf{z}^{(l)} &= \text{MSA}\left(\mathbf{x}^{(l)}\right) + \mathbf{x}^{(l)}, \\ \mathbf{x}^{(l+1)} &= \text{FFN}\left(\mathbf{z}^{(l)}\right) + \mathbf{z}^{(l)}, \end{aligned} \quad (1)$$

where MSA is used to capture the global dependencies among the patches, while the FFN is used to model the local dependencies within each patch.

For enabling knowledge transfer from computationally expensive and large models to smaller ones, we propose a multiplex distillation method for acquiring a light-SAM image backbone. We first obtain a large SAM model pre-trained on a large-scale dataset, e.g., SA-1B[18], then we distill the knowledge from the large model to a smaller one. The multiplex distillation process is conducted by diminishing the multiplex relation map gap $\mathcal{L}_{md}$ among teacher and student models' intermediate layers' features denoted as $F_T \in \mathbb{R}^{N \times D_T}$ and $F_S \in \mathbb{R}^{N \times D_S}$ respectively and minimizing the KL divergence $\mathcal{L}_{kl}$ between the output of the teacher model $p_s(\mathbf{x})$ and the output of student model $p_t(\mathbf{x})$. The distillation loss is defined as follows:

$$\begin{aligned} \mathcal{L}_{kl} &= \tau^2 \mathcal{F}_{KL}\left(p_s(\mathbf{x})/\tau, p_t(\mathbf{x})/\tau\right), \\ \mathcal{L}_{md} &= \left\| F_S F_S^\top - F_T F_T^\top \right\|_F^2, \end{aligned} \quad (2)$$

where $\mathcal{F}_{KL}$ denote the KL divergence loss, with $\tau$ as the temperature parameter. Hence, the overall loss function for distillation is:

$$\mathcal{L} = \mathcal{L}_{kl} + \sum_l \mathcal{L}_{md}. \quad (3)$$

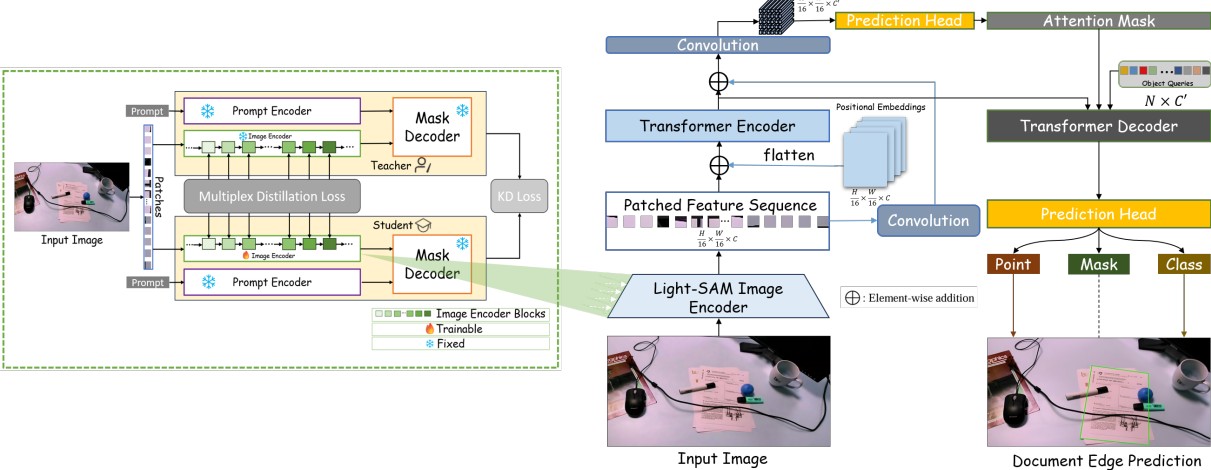

**Figure 2: The pipeline of RDLNet(normalization are omitted for simplicity). RDLNet adopts an encoder-decoder structure with a light-SAM ViT backbone distilled from a larger model. The input RGB image $I \in \mathbb{R}^{H \times W \times 3}$ is patched into image block sequence $I' \in \mathbb{R}^{\frac{H}{16} \times \frac{W}{16} \times C}$ utilizing convolutional operation with both the kernel size and the stride of 16. Attached with positional embedding, the patched feature sequence feeds into the transformer encoder for spatial dependencies capturing. In decoding object queries[6], we introduce masked attention based upon prior predicted mask for faster training convergence rate and better performance and output final predictions exploiting point, mask and class embedding.**

### 3.2 Cross-Level Encoder

To fully harness information from the backbone, we design a cross-level encoder that utilizes the low-level to high-level features from backbone intermediate layers. They are projected into a new feature space and concatenated to form the feature map $I'_{\text{proj}}$

$$I'_{\text{proj}} = \text{Concat}(\text{Conv}(I'_1), \text{Conv}(I'_{l/2}), \text{Conv}(I'_l)), \quad (4)$$

where Conv denotes the convolutional operation, $I'_1$, $I'_{l/2}$ and $I'_l$ are the feature maps of the first, middle and last layer of the backbone output respectively. For spatial dependencies capturing, we take advantage of deformable attention[41] to guide the encoding process with important regions from the image and bridge visual tokens' relations more effectively.

Enriched with positional information via positional embedding $I'_{\text{proj}} \leftarrow I'_{\text{proj}} + P_e, P_e \in \mathbb{R}^{\frac{H}{16} \times \frac{W}{16} \times C}$, the input for each encoder layer conduct:

$$\tilde{I}'_{\text{proj}} = \text{DMSA}(I'_{\text{proj}}) + I'_{\text{proj}},$$
$$I'_{\text{proj}} = \text{FFN}(\tilde{I}'_{\text{proj}}) + \tilde{I}'_{\text{proj}}, \quad (5)$$

where DMSA denotes the deformable multi-head self-attention. To prepare input mask for decoder attention mechanism and filter out background objects, we also project the last layer output from the backbone with convolutional operation and obtain prior mask $M_p \in \mathbb{R}^{N_o \times \frac{HW}{16}}$ with mask embedding $\psi$ and object queries $Q \in \mathbb{R}^{N_o \times C'}$:

$$M_p = \text{Reshape}\left(\text{Flatten}\left(\text{Conv}(I'_l) + \phi(I'_l)\right) \cdot \psi\left(Q^\top\right)\right), \quad (6)$$

where $\phi$ and $N_o$ denote the encoding operation and the number of object queries. Reshape and Flatten are reshaping ($HW/16 \rightarrow$

$H/16 \times W/16$) and flattening ($H/16 \times W/16 \rightarrow HW/16$) operations respectively.

### 3.3 Masked Decoder

Recent studies[9, 33] suggest global context features eventuate in the slow convergence of Transformer-based models, as it takes numerous training epochs for attention mechanism to learn and attend to local foreground regions. Thus, we propose a masked decoder and mask some of the background areas based upon prior mask prediction with Eq.6 to filter global context features for attention operations as shown in Figure 3. With masked cross attention, we compute the result matrix via:

$$\mathbf{x}^{(l)} = \text{softmax}\left(M'^{(l-1)}_p + \mathbf{Q}^{(l)}\mathbf{K}^{(l)\top}\right)\mathbf{V}^{(l)} + \mathbf{x}^{(l-1)}. \quad (7)$$

Here, $M'^{(l-1)}_p$ is the previous reshaped and repeated prior mask $M_p$ for decoder layer $l$, each decoder layer exploits mask prediction from the last layer while the first decoder layer utilizes mask acquired from Eq.6 to initialize $M'^{(0)}_p$. $\mathbf{Q}^{(l)}$, $\mathbf{K}^{(l)}$ and $\mathbf{V}^{(l)}$ are the query, key and value matrices respectively. $\mathbf{x}^{(l-1)}$ is the output of the previous decoder layer. The object queries after decoding are then fed into the triple embedding branches of point, mask and class for prediction. In particular, class embedding branch endues our model with the ability to distinguish among different types of documents on the instance level, e.g., top document and overlapped document or document with different categories such as paper and magazine etc. Contrasting to conventional methods like [8, 14, 16] that demand heavy post-processing, e.g., polygon approximation, to attain final document border or corner points predictions, RDLNet directly output polygon corner points with its point embedding. In the case

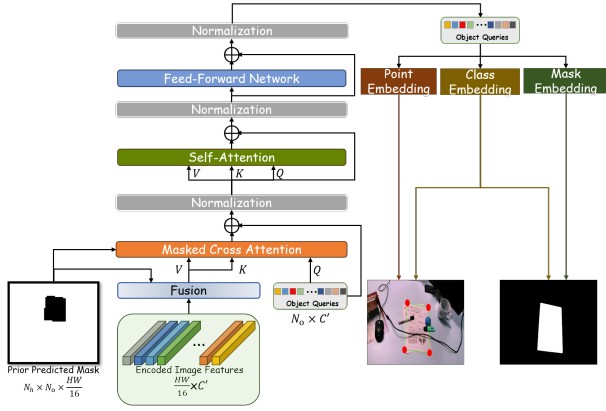

**Figure 3: Masked Decoder overview(positional embedding and predictions from intermediate decoder layers are omitted in this figure for simplicity). After acquiring prior predicted mask $M_p$, we fuse it with encoded backbone features from the encoder via masking the background area and repeat it to $M_p' \in \mathbb{R}^{N_h \times N_o \times \frac{HW}{16}}$ where $N_h$ is the number of heads for multi-head masked attention operation. Note that we switch the order of self-attention and cross-attention so that local areas can be attended solely in both attention operations for faster training convergence rate and performance.**

of documents with irregular shapes or complex document targets, we also make use of the mask embedding branch as an auxiliary approach for a precise approximation on their edges.

## 3.4 Loss Functions

To supervise the training of RDLNet, $N_o$ decoded object queries are used to generate a set of $N_o$ predictions $\{\hat{y}_i\}_{i=1}^{N_o}$ via point, mask and class embedding consisting of point coordinates $\hat{q}$, raw mask logits $\hat{m}$ and class probabilities $\hat{p}_i(c)$. Following [6], the optimal permutation $\{\hat{y}_{\sigma i}\}_{i=1}^{N}$ is determined by bipartite matching between the ground truth and the predictions.

In lieu of using focal loss[22], we use the binary cross-entropy loss:

$$\mathcal{L}_{\text{mask}} = -\frac{1}{N_o} \sum_{i=1}^{N} \left( m_i \log\left(\hat{m}_{\sigma(i)}\right) + (1 - m_i) \log\left(1 - \hat{m}_{\sigma(i)}\right) \right), \tag{8}$$

and dice loss [25]:

$$\mathcal{L}_{\text{dice}} = 1 - \frac{2 \times \sum_{i=1}^{N_o} \left( m_i \times \hat{m}_{\sigma(i)} \right)}{\sum_{i=1}^{N_o} m_i + \sum_{i=1}^{N_o} \hat{m}_{\sigma(i)}}, \tag{9}$$

for our mask loss. Besides, we propose a distance loss for point branch prediction supervision,

$$\mathcal{L}_{\text{distance}} = \frac{1}{N_o} \sum_{i=1}^{N_o} \left\| \hat{q}_{\sigma(i)} - q \right\|, \tag{10}$$

while foreground document class predictions are supervised via standard cross entropy:

$$\mathcal{L}_{\text{cls}} = \sum_{i=1}^{N_o} \text{CrossEntropy}\left( c_i, \hat{p}_{\sigma(i)}(c) \right). \tag{11}$$

Ultimately, the total loss is composed of the weighted sum of the above losses:

$$\mathcal{L}_{\text{total}} = \lambda_1 \mathcal{L}_{\text{cls}} + \lambda_2 \mathcal{L}_{\text{distance}} + \lambda_3 \mathcal{L}_{\text{dice}} + \lambda_4 \mathcal{L}_{\text{mask}}, \tag{12}$$

with $\lambda_i$ as adjustable hyper-parameters.

## 3.5 Implementation Details

We train RDLNet with AdamW optimizer with weight decay set to $10^{-4}$ under the initial learning rate of $10^{-4}$ with a learning rate drop by a factor of $10^{-1}$ every 40000 iterations. Experiments are carried out on 1 NVIDIA A800 GPUs with the mini-batch size of 8 for 160000 training iterations in total. Loss weights are empirically set to $\lambda_1 = 2$, $\lambda_2 = 2$, $\lambda_3 = 5$ and $\lambda_4 = 5$. The input images are resized to 1024 pixels for both width and height.

## 4 RWMD DATASET

**Table 1: Document categories and corresponding image quantity of RWMD dataset**

| Category | Quantity |
|---|---|
| printed paper document | 355 |
| book | 360 |
| test paper | 172 |
| students' workbook | 204 |
| receipt | 246 |
| card | 223 |
| exercises book | 164 |
| certificate | 141 |
| other multi-categories | 144 |
| total | 2009 |

To the best of our knowledge, there is no existing publicly available dataset comprising of documents captured by smartphones in real conditions with various types of document and complex scenarios. We create such a comprehensive benchmark dataset RWMD for document localization. The dataset will soon be made public, and we hope that it will serve the scientific community for more document localization methods to boom. The images in this dataset are captured by eight different mobile phones including iPhone13, Samsung Galaxy S21 Ultra, HUAWEI nova 7 SE, HUAWEI P30, HUAWEI P50 Pro, VIVO S17e, VIVO X21A, and Xiaomi 8. The RWMD dataset contains a total of 2009 images which are divided into nine categories. Moreover, each category contains documents in different styles with different real background, distortion, rotation, perspective transformation, contrast, light intensity, occlusion and other real situations. The samples of different types of documents with various scenarios are shown in Figure 4. These document pictures taken in the real world contain a variety of real scenes, which pose great challenges to the document localization algorithm. In order to meet the needs of different methods, the RWMD annotation

documents contain a variety of information, including masks of different document areas, class labels of different document instances corresponding to the masks, and corner points coordinates of the document. If the image contains multiple different documents, in order to be closer to the actual usage scenarios of the document localization algorithms on the devices, we distinguish the main document and the background document based on the area size and position of different documents in the image. As presented in Figure 5, different document areas are marked separately, the main document is given the maximum class label value 2 and the corner points coordinates are also marked.

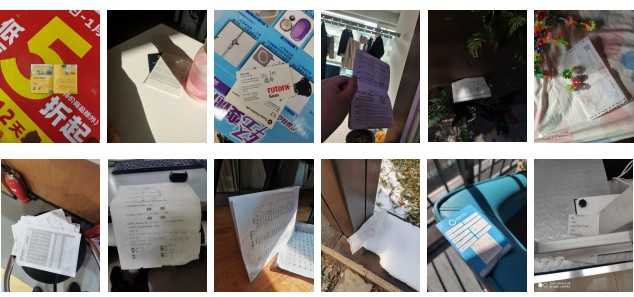

**Figure 4: The sample images of RWMD dataset.**

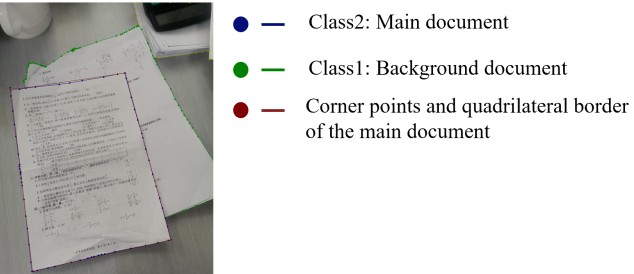

**Figure 5: Annotation example of RWMD.**

## 5 EXPERIMENT

In this section, we evaluate the performance of our proposed method on five datasets: SmartDoc dataset [4], Extended smartdoc dataset [10], CDPhotoDataset [11], MIDV2020 dataset [3], and RWMD dataset. The existing four datasets ( SmartDoc dataset , Extended smartdoc dataset, CDPhotoDataset, and MIDV2020 dataset) have already been described in the introduction section and RWMD dataset is introduced in the previous section. They are all divided into 75% for training and 25% for testing, separately. To illustrate the effectiveness of our method, we compare it with state-of-the-art document localization approaches from several aspects. Furthermore, we evaluate our proposed design modules through ablations on the RWMD dataset.

### 5.1 Evaluation Metric

To quantitative evaluate the accuracy of the proposed model, we compute the Jaccard index (JI), which is described in the SmartDoc competition report [4] and adopted in other works [8] [34]. Firstly, we remove the perspective transformation of the ground-truth area and the predicted area to get their correct situations $D_{gt}$ and $D_p$. Then for each frame or image, the Jaccard index (JI) is calculated as follows:

$$JI(f) = \frac{area(D_{gt}) \cap area(D_p)}{area(D_{gt}) \cup area(D_p)}. \tag{13}$$

### 5.2 Comparison with SOTA

*5.2.1 Experiment 1: Evaluation performance on SmartDoc dataset.* The first experiment compares the proposed method with other state-of-art (SOTA) algorithms in the SmartDoc dataset. The images in the dataset can be divided into five categories(BG01-BG05) according to different backgrounds. Only BG05 is complex, with similar overlapping document and strong occlusions from other objects. As shown in Table 2, our RDLNet outperforms the previous works on all sub-test scenarios from BG1 to BG5. The overall JI of RDLNet achieves 99.53% and exceeds other methods including HU-PageScan in [8], which had achieved the highest overall JI 99.23% before. At the same time, we can observe that our method can achieve almost the same accuracy as BG1-BG4 on BG5, which proves the robustness of our method in more complex scenarios.

*5.2.2 Experiment 2: Evaluation performance on CDPhotoDataset and Extended SmartDoc dataset.* In the second experiment, we evaluate our model and the SOTA methods performance on CDPhoto-Dataset and Extended SmartDoc dataset. The experimental results of the two datasets are presented in Table 3, which shows that our method still obtains the best accuracy performance when comparing with the other previous methods. In particular, although LDRNet achieves good results on the SmartDoc dataset, it is not suitable for the CDPhotoDataset. Because CDPhotoDataset contains multiple documents in one image that need to be detected, however, LDRNet can only detect a single quadrilateral document. Additionally, in order to compare the efficiency of different methods, we also compared the inference speed of our method with others. As Table 3 shows, our method exhibits an average speed of 0.0385s and 0.0408s per image on the CDPhotoDataset and Extended SmartDoc datasets, respectively, which is comparable to other fastest methods. All the experiment results demonstrate that our proposed RDLNet has great capabilities for locating documents in natural scene images accurately and efficiently.

*5.2.3 Experiment 3: Evaluation performance on RWMD dataset.* Although many preceding methods has obtained outstanding detection metric results with high JI on publicly available document datasets mentioned above, they are not capable of coping with problems of the real-world document detection and generalizing well, since the scenarios in the datasets above are relatively deprived in the diversity of background and don't contain sufficient complex document photography scenes existing in the real situations. In the third experiment, we evaluate our method and other SOTA algorithms on our new dataset RWMD. RWMD contains a rich

**Table 2: JI Results on SmartDoc Dataset**

| Method | Background(BG) | | | | | Average |
|---|---|---|---|---|---|---|
| | BG01 | BG02 | BG03 | BG04 | BG05 | |
| HU-PageScan[8] | / | / | / | / | / | 0.9923 |
| LDRNet[34] | 0.9877 | 0.9838 | 0.9862 | 0.9802 | 0.9858 | 0.9849 |
| SEECS-NUST-2[16] | 0.9832 | 0.9724 | 0.983 | 0.9695 | 0.9478 | 0.9743 |
| LRDE[5] | 0.9869 | 0.9775 | 0.9889 | 0.9837 | 0.8613 | 0.9716 |
| SmartEngines[5] | 0.9885 | 0.9833 | 0.9897 | 0.9785 | 0.6884 | 0.9548 |
| OctHU-PageScan[8] | / | / | / | / | / | 0.9237 |
| Mask R-CNN[14] | / | / | / | / | / | 0.9063 |
| NetEase[5] | 0.9624 | 0.9552 | 0.9621 | 0.9511 | 0.2218 | 0.8820 |
| RPPDI-UPE[5] | 0.8274 | 0.9104 | 0.9697 | 0.3649 | 0.2163 | 0.7408 |
| SEECS-NUST[5] | 0.8875 | 0.8264 | 0.7832 | 0.7811 | 0.0113 | 0.7393 |
| **RDLNet(Ours)** | **0.9962** | **0.9952** | **0.9954** | **0.9946** | **0.9946** | **0.9953** |

**Table 3: JI Results on CDPhotoDataset(CDP) and Extended SmartDoc Dataset(ESD)**

| Method | CDP | | ESD | |
|---|---|---|---|---|
| | JI | time | JI | time |
| HU-PageScan[9] | 0.9896 | 0.0422 | 0.9908 | 0.1067 |
| U-net[28] | 0.9916 | 0.0799 | 0.9904 | 0.2551 |
| LDRNet[34] | / | / | 0.9315 | 0.0171 |
| Mask R-CNN[14] | 0.9498 | 0.1176 | 0.924 | 0.3688 |
| OctHU-PageScan[8] | 0.9456 | 0.0232 | 0.7684 | 0.0231 |
| GOP[19] | 0.6868 | 0.0597 | 0.7913 | 0.1394 |
| **RDLNet(ours)** | **0.9905** | 0.0385 | **0.9945** | 0.0408 |

**Table 4: JI Results on our new dataset RWMD**

| Method | JI |
|---|---|
| Mask R-CNN[14] | 0.8622 |
| HU-PageScan[9] | 0.7219 |
| OctHU-PageScan[8] | 0.6821 |
| Efficient-SAM[38] | 0.6726 |
| SAM[18] | 0.6988 |
| U-net[28] | 0.6380 |
| LDRNet[34] | 0.5261 |
| SEECS-NUST-2[16] | 0.3751 |
| **RDLNet(ours)** | **0.9123** |

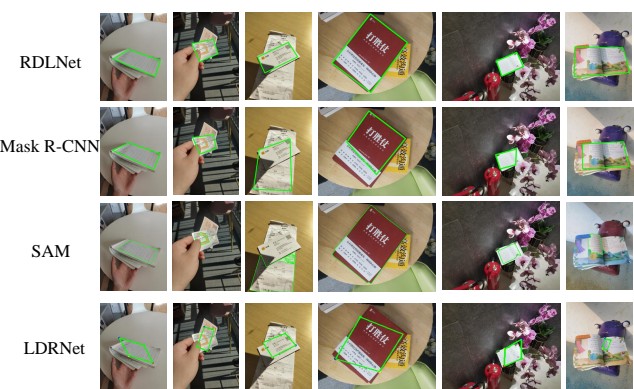

**Figure 6: Results comparison of different methods on RWMD test dataset**

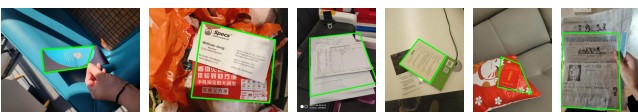

**Figure 7: Failed result samples of our method on RWMD test dataset**

variety of common document types and real-world scenes, which is much more complex and better to reflect the real document scenarios. As described in Table 4 , our proposed method achieves the highest JI value and outperforms other methods. Notably, we also directly apply pre-trained Efficient-SAM[38] and SAM[18], which are prompt-based segmentation methods, on the RWMD test dataset for document localization with a single center point prompt as prompt input and post-processed output masks as final predictions. It turns out SAM-series models don't support generic document segmentation and localization well due to its specific characteristics of prompt design. As a result, we acquired relatively low JI from SAM models. Figure 6 shows the examples of the comparison results between our method, maskrcnn, LDRnet and SAM on the RWMD test dataset. These results present that our proposed method can handle the more complex document localization tasks and it exhibits better robustness in real complex scenarios.But at the

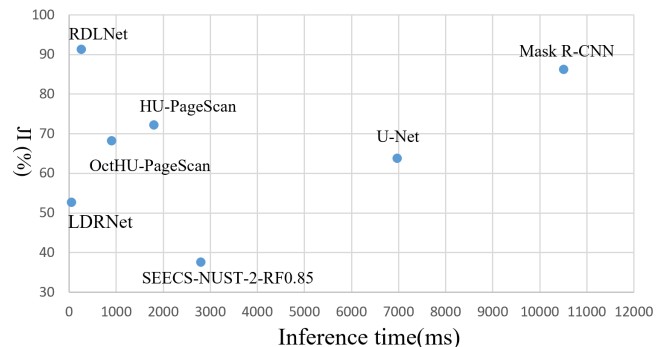

**Figure 8: Comparison of inference accuracy and speed**

same time, we can also see that although our method has obtained a relatively high JI value, there is still a certain space for further improvement. This also shows that many existing document localization problems have not been well solved. Figure 7 shows some document localization results those with still problems. The area within the green line is the detected document area. The results also verify the significance of the RWMD dataset we construct in this paper for document localization methods measurement, and provide a more realistic and challenging benchmark dataset for future research.

*5.2.4 Experiment 4: Evaluation of inference speed.* Furthermore, in order to verify the efficiency of algorithms on mobile devices, we compare the test duration of different methods on mobile phone HUAWEI Mate 40. In Figure 8, the vertical axis is the JI of the model while the horizontal axis is the inference time. Figure 8 illustrates that our proposed model RDLNet obtains the highest accuracy with low inference time. As can be seen from Figure 8 the inference speed of our model is second only to LDRNet, but the accuracy is much higher than LDRNet. Mask R-CNN obtains the accuracy second to our model, but the inference efficiency is far lower than our method.

**Table 5: JI Results on MIDV2020 dataset**

| Method | Fine-tune | JI |
|---|---|---|
| Mask R-CNN[14] | ✗ | 0.8607 |
| HU-PageScan[9] | ✗ | 0.6735 |
| OctHU-PageScan[8] | ✗ | 0.7154 |
| LDRNet[34] | ✗ | 0.4221 |
| U-net[28] | ✗ | 0.3201 |
| SEECS-NUST-2[16] | ✗ | 0.3690 |
| **RDLNet(ours)** | ✗ | **0.9266** |

*5.2.5 Experiment 5: Generalization on unseen documents and background.* To verify our model's generalization ability to new documents on unseen background, we cross-validate our method on the dataset which is not used for training. Specifically, we use the RWMD training set to train models and MIDV2020 test set to verify the detection effects. As demonstrated in Table 5, our method is

significantly better than other methods on unseen dataset. Especially, the accuracy of our method outperforms the second-place method Mask R-CNN by 15.04%. The results prove that our method has good generalization capability on unseen scenarios.

## 5.3 Ablation Study

**Table 6: JI Results With Different Modules on RWMD dataset**

| Baseline | Cross Level | Distillation | Point Branch | JI |
|---|---|---|---|---|
| ✔ | | | | 0.8031 |
| ✔ | ✔ | | | 0.8683 |
| ✔ | ✔ | ✔ | | 0.9014 |
| ✔ | ✔ | | ✔ | 0.8713 |
| ✔ | | ✔ | | 0.8813 |
| ✔ | ✔ | ✔ | ✔ | 0.9123 |

In our model, we propose several novel modules, such as cross-level encoder, multiplex distillation[1], and point-mask joint supervision. To verify the effectiveness of these modules, we carry out corresponding ablation experiments. Table 6 shows the verification results, from which we can observe that the proposed modules are essential for accuracy.

## 6 CONCLUSION

In this paper, we introduced RDLNet, a novel model designed for robust document localization in real-world images. Leveraging a light-SAM backbone, RDLNet efficiently extracts hierarchical visual features and employs a decoder that harnesses mask attention and object queries to accurately predict document corner points, segmentation masks, and categories. The RDLNet exploits the light-SAM backbone to extract the cross-level vision features of the document images and jointly predicts the corner points coordinates, area mask and class of the documents following decoder based on masked attention and object query. At the same time, we create a new document localization dataset RWMD to fill the poverty of dataset for multiple document types and complex scenarios in the wild. Extensive experimental results have demonstrated that our model achieved the state of the art accuracy, while using low inference time. Additionally, the experiments on unseen documents and backgrounds highlight RDLNet's strong generalization capability, underscoring its practical applicability in diverse settings. We hope this research not only advances the state-of-the-art in document localization but also establishes a challenging benchmark that will likely drive future progress in the field.

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
