# OpenReview forum: "RDLNet: A Novel and Accurate Real-world Document Localization Method"
_acmmm.org/ACMMM/2024/Conference — MM2024 Poster_

### Official Review · Reviewer_PgqY · 2024-05-17

**Rating:** 3
**Confidence:** 4

**Summary:**

This paper proposed a real-world document localization method and a new dataset for a real-world document localization dataset. The proposed method consists of a light-SAM encoder and a masked attention decoder. A triple-branch predictions that contains corner point coordinates, instance-level segmentation area and categories of different documents, are adopted to accomplish the task. The experimental results on mulitple benchmarks show the effectiveness of the proposed method.

**Strengths:**

Dataset Contribution: The creation of a new and comprehensive dataset is a significant contribution. It addresses the existing gap in the field by providing a diverse set of document types and real-world scenarios, which is essential for training and evaluating robust localization models.


State-of-the-Art Performance: The method's performance on multiple benchmarks is commendable. Achieving state-of-the-art (SOTA) results in terms of both accuracy and inference speed is a testament to the effectiveness of the proposed approach.

**Limitations:**

Technical Novelty: While the paper presents a new method, the core components such as the light-SAM encoder and the masked attention decoder are not novel in themselves. This raises questions about the extent of the innovation in the proposed method.


Multi-Task Learning: The triple-branch prediction can be seen as an extension of multi-task learning, which is not a new concept. The paper could benefit from a deeper exploration of how this approach differs from or improves upon existing multi-task learning frameworks.


Fairness of Comparisons: The paper's claims of superiority over other methods are contingent on the fairness of the comparisons made. It is crucial to understand whether the compared methods were evaluated under the same conditions, including the use of identical training datasets. Without this information, the comparative analysis may not be entirely reliable.

**Suitability:**

2

---

### Official Review · Reviewer_Xhro · 2024-05-20

**Rating:** 5
**Confidence:** 4

**Summary:**

The increasing use of smartphones for capturing documents in various real-world conditions has underscored the need for robust document localization technologies. Current challenges in this domain include handling diverse document types, complex backgrounds, and varying photographic conditions such as low contrast and occlusion. However, there currently are no publicly available datasets containing these complex scenarios and few methods demonstrate their capabilities on these complex scenes. To address the issues, the paper proposes a document localization method named RDLNet, which consists of light-SAM backbone multiplex distillation, cross-level encoder, and masked decoder. Besides, a large-scale real-world document localization dataset is proposed, which reflects more complex scenarios of document images in the wild. Experiments are performed on five datasets to demonstrate the effectiveness of the proposed method.

**Strengths:**

1) The proposed dataset includes documents captured by smartphones in real conditions with various types of documents and complex scenarios, which is valuable to the community.
2) The proposed RDLNet not only utilizes the knowledge from SAM but also produces various types of predictions with unified queries.
3) The experiments are well conducted and the paper is overall written.

**Limitations:**

1) The details are absent in Section 3, e.g., how the prompt is utilized for Light-SAM Backbone Multiplex Distillation.
2) Typos. L109: Real world -> Real-world. L433: HW/16 -> HW/16^2.

**Suitability:**

3

---

### Official Review · Reviewer_iwnm · 2024-05-27

**Rating:** 3
**Confidence:** 3

**Summary:**

This paper introduces a new method for real-world document localization called RDLNet and establishes a new benchmark dataset RWMD. The innovation of this article mainly lies in the design of a lightweight SAM encoder, a decoder based on masked attention and object queries, and the proposal of a three-branch prediction method. These methods perform exceptionally well in multiple benchmark tests, especially in document localization tasks within complex scenes.

**Strengths:**

1. The authors propose a lightweight SAM encoder and a decoder based on masked attention, combined with a three-branch prediction method, to address multiple challenges in document localization.
2. The RWMD dataset contains a rich variety of document types and complex shooting scenes, providing valuable resources for future research.
3. The experimental results in the paper demonstrate the superior performance of RDLNet across multiple benchmark tests, particularly in robustness within complex scenes.

**Limitations:**

1. While RDLNet shows significant improvement in accuracy, further evaluation is needed regarding the algorithm's complexity and inference time performance on mobile devices. It is suggested that the authors provide more detailed data on inference time and resource consumption.
2. The paper mentions the use of multiple distillation methods to obtain a lightweight SAM encoder. It is recommended to provide additional details regarding the distillation process and the selection of hyperparameters.
3. When describing the RWMD dataset, it is advisable to provide more information about the methods for data annotation consistency and quality control to ensure the dataset's high quality and reliability.
4. While RDLNet performs excellently in this paper, its performance on larger-scale datasets still needs validation. It is suggested that the authors provide more discussion and experiments on the algorithm's scalability.

**Suitability:**

2

---

### Official Review · Reviewer_2Qcn · 2024-06-25

**Rating:** 5
**Confidence:** 2

**Summary:**

The paper addresses the increasing need for robust document localization methods due to the widespread use of smartphones for capturing documents. The authors highlight the lack of open-source datasets that cover complex real-world scenarios. To address this gap, they created the comprehensive Real-world Mobile Document (RWMD) benchmark dataset, which covers 9 categories and contains a total of 2009 images. Additionally, the paper proposes a novel Real-world Document Localization Network (RDLNet), which features a light-SAM encoder and a masked attention decoder, aiming to transfer the generalization capability of SAM to document localization. The method claims to outperform state-of-the-art approaches on multiple benchmarks.

**Strengths:**

The paper introduces a novel triple-branch structure embedded in the proposed RDLNet, which simultaneously predicts corner points, area masks, and document categories, enhancing the accuracy and efficiency of document localization.

The authors conducted experiments and comparisons with other methods, demonstrating the effectiveness and robustness of the proposed RDLNet.

The authors constructed the Real-world Mobile Document (RWMD) dataset, which includes various document types and challenging real-world scenarios.

**Limitations:**

When presenting table values, it is common to list methods chronologically or in ascending order of results. The authors' table values are in reverse order. To indicate ranking order, it is recommended to re-order or include a ranking number in the first column.

In document localization, images are often captured casually, resulting in blurriness, low lighting, low contrast, or occlusion. Not only boundary-detection methods need traditional image processing. Smartphones recording metadata, like ISO, image size, lens type, and shutter speed, may contribute to improving localization accuracy.

**Suitability:**

3

---

### Meta-Review · Area_Chair_R5Yg · 2024-07-01

**Recommendation:** Accept (Poster)
**Confidence:** 4

**Metareview:**

Summary:

The paper addresses the increasing need for robust document localization methods due to the widespread use of smartphones for capturing documents. The authors propose a novel Real-world Document Localization Network (RDLNet) featuring a light-SAM encoder and a masked attention decoder, aiming to improve the generalization capability of document localization tasks. Additionally, the paper introduces the Real-world Mobile Document (RWMD) benchmark dataset, which covers 9 categories and contains 2009 images, demonstrating the effectiveness of RDLNet through extensive experiments on multiple benchmarks.

Strengths:
1. RDLNet introduces a novel triple-branch structure that simultaneously predicts corner points, area masks, and document categories, enhancing localization accuracy and efficiency.
2. The creation of the RWMD dataset, which includes various document types and challenging real-world scenarios.
3. Experimental results show that RDLNet outperforms state-of-the-art methods on multiple benchmarks, particularly in complex real-world scenes.

Limitations:
1. While the method achieves impressive results, the core components like the light-SAM encoder and masked attention decoder are not novel, which may limit the perceived innovation.
2. More details on data annotation consistency and quality control for the RWMD dataset are needed. Additionally, the scalability of RDLNet on larger-scale datasets requires further validation.
3. The framework's complexity, involving split learning, knowledge distillation, and encryption, might complicate implementation and maintenance. Furthermore, details about the distillation process and selection of hyperparameters should be provided.
4. The comparative analysis may not be entirely reliable without ensuring that all methods were evaluated under the same conditions, including the use of identical training datasets.

According to the comments, the final average rating leans to the positive side. Therefore, I will recommend this paper as Accept (Poster).